# Vaccination with a Protective Ipa Protein-Containing Nanoemulsion Differentially Alters the Transcriptomic Profiles of Young and Elderly Mice following *Shigella* Infection

**DOI:** 10.3390/vaccines12060618

**Published:** 2024-06-04

**Authors:** Ti Lu, Murugesan Raju, Debaki R. Howlader, Zackary K. Dietz, Sean K. Whittier, David J. Varisco, Robert K. Ernst, Lyndon M. Coghill, William D. Picking, Wendy L. Picking

**Affiliations:** 1Bond Life Sciences Center and Department of Veterinary Pathobiology, University of Missouri, Columbia, MO 65211, USA; drhb7r@missouri.edu (D.R.H.); pickingw@missouri.edu (W.D.P.); 2Bioinformatics and Analytic Core, University of Missouri, Columbia, MO 65211, USAlcoghill@missouri.edu (L.M.C.); 3MU Institute for Data Science and Informatics, University of Missouri, Columbia, MO 65211, USA; 4Department of Microbial Pathogenesis, University of Maryland, Baltimore, MD 21201, USA

**Keywords:** *Shigella* spp., type III secretion system, vaccine, aging immunity, dual RNA-seq

## Abstract

*Shigella* spp. are responsible for bacillary dysentery or shigellosis transmitted via the fecal–oral route, causing significant morbidity and mortality, especially among vulnerable populations. There are currently no licensed *Shigella* vaccines. *Shigella* spp. use a type III secretion system (T3SS) to invade host cells. We have shown that L-DBF, a recombinant fusion of the T3SS needle tip (IpaD) and translocator (IpaB) proteins with the LTA1 subunit of enterotoxigenic *E. coli* labile toxin, is broadly protective against *Shigella* spp. challenge in a mouse lethal pulmonary model. Here, we assessed the effect of LDBF, formulated with a unique TLR4 agonist called BECC470 in an oil-in-water emulsion (ME), on the murine immune response in a high-risk population (young and elderly) in response to *Shigella* challenge. Dual RNA Sequencing captured the transcriptome during *Shigella* infection in vaccinated and unvaccinated mice. Both age groups were protected by the L-DBF formulation, while younger vaccinated mice exhibited more adaptive immune response gene patterns. This preliminary study provides a step toward identifying the gene expression patterns and regulatory pathways responsible for a protective immune response against *Shigella*. Furthermore, this study provides a measure of the challenges that need to be addressed when immunizing an aging population.

## 1. Introduction

*Shigella* is an intracellular pathogen that causes severe diarrhea or dysentery, with 270 million cases globally each year resulting in approximately 212,000 deaths [1]. *Shigella* spp. include *S. dysenteriae*, *S. flexneri*, *S. boydii*, and *S. sonnei*, which are further divided into more than 50 serotypes based on O-antigen composition [1]. *S. flexneri* is a significant public health concern, particularly in low- and middle-income countries with poor sanitation and hygiene practices [1]. Treatment of *S. flexneri* infections typically involves antibiotics, such as fluoroquinolones, β-lactams, and cephalosporins, but the emergence of antibiotic-resistant strains has become a growing problem [2,3]. Therefore, vaccine development to combat *Shigella* spp. is an unmet, but highly important, public health need. Currently, there is no approved vaccine for *Shigella* infection. While several vaccine candidates, including inactivated cell vaccines and live-attenuated vaccines, are undergoing clinical trials, their widespread adoption is hindered by challenges such as low immunogenicity, a lack of cross-protection, stringent storage requirements, and the potential risk of functional mutations [4].

To address these challenges, a considerable research effort been directed toward subunit vaccines, with a particular emphasis on those composed of proteins derived from the type III secretion system (T3SS) that are under development [5]. The T3SS is a specialized apparatus employed by many pathogenic bacteria, such as *Shigella*, to directly inject effector proteins into host cells [6]. The T3SS tip protein, IpaD, and the translocator, IpaB, are both highly conserved across all *Shigella* serotypes and are required for infection, rendering them strong candidates for developing a serotype-independent subunit vaccine [7]. We have previously shown that the intranasal (IN) administration of these two proteins with the mucosal adjuvant dmLT (double mutant heat-labile enterotoxin from enterotoxigenic *E. coli*) induces cross-protection against both *S. flexneri* and *S. sonnei* [8]. To eliminate the need for purifying multiple protein components, we genetically fused LTA-1, the enzymatic moiety of dmLT, with a genetic fusion of IpaB and IpaD. This fusion, known as L-DBF, demonstrated enhanced broad protection against lethal *Shigella* spp. challenges [9]. Moreover, we showed that L-DBF, with a newly developed TLR4 agonist called BECC438 (a detoxified lipid A analogue identified as Bacterial Enzymatic Combinatorial Chemistry candidate #438), formulated in an oil-in-water emulsion, has a very high protective efficacy at low antigen doses against lethal challenge [10].

The mouse pulmonary model for *Shigella* research is efficient, offering valuable insights into *Shigella* pathogenesis and host responses [1,11]. While mouse intestinal models exist, it is crucial to acknowledge their limited commercial availability, especially for aging studies [12]. In contrast, the mouse *Shigella* pulmonary challenge model remains widely used and accessible, serving as a vital tool for studying infection dynamics and evaluating vaccine efficacy [1,13]. This model facilitates controlled investigations of mucosal immune responses and enables early-stage assessment of candidate vaccines’ effectiveness in preventing *Shigella* infections [1,13]. Despite the availability of intestinal mouse models, the ongoing use of the *Shigella* pulmonary challenge model underscores its continued relevance and significance in preclinical research for vaccine development against *Shigella* infection.

While shigellosis affects individuals of all age groups and is an important cause of morbidity and mortality in children under the age of five [14,15], we focus here on understanding its impact on both young and elderly populations [16]. The elderly, often defined as individuals aged 65 years and older, are known to experience an increased susceptibility to infectious diseases due to age-related changes in their immune system and underlying health conditions [17]. Prior research has shown that aging leads to a decrease in naive T cells and antibody diversity, while increasing memory T cells and NK cells with a diminished function [18,19]. Macrophages and dendritic cells exhibit reduced phagocytosis and antigen presentation capabilities. Increased pro-inflammatory cytokines also contribute to chronic inflammation in elderly mice [20,21]. Moreover, the differential expression of genes in aged immune cells has been seen, which impacts the overall immune function and the body’s response to infection [21]. In the case of *Shigella* spp., this susceptibility can lead to more severe symptoms and complications in elderly individuals, making it an important area of research and a public health concern [16]. Thus, the development of a vaccine specifically tailored for the elderly against *Shigella* spp. infections holds immense importance.

In this study, we used L-DBF with BECC438 or BECC470 (Bacterial Enzymatic Combinatorial Chemistry Candidate #438 or #470) in an oil-in-water emulsion (ME) to determine this formulation’s effectiveness in protecting against *Shigella* pulmonary challenge in young and elderly mice, and we evaluated the effects of this vaccine formulation on cellular and humoral immune responses. Additionally, a lung bulk mRNA seq analysis was also performed on young and elderly mice to determine the immune pathways activated by the vaccine and how they compare to those activated by *Shigella* infection in non-immunized mice in these different age groups. Understanding the differential immune responses at transcriptomic levels between young and elderly populations will be informative in the development of effective vaccines tailored to the specific needs of each demographic, thereby addressing the increased susceptibility and severity of *Shigella* infections in elderly individuals and ultimately improving public health outcomes.

## 2. Materials and Methods

### 2.1. Materials

BECC470 and BECC438 were prepared by extraction from *Yersinia pestis* after introducing lipid A-modifying enzymes. They have been previously shown to be nontoxic in rabbits [22,23]. Squalene was purchased from Echelon Biosciences (Salt Lake City, UT, USA). Chromatography columns were from GE Healthcare (Piscataway, NJ, USA). All other reagents were from Sigma (St. Louis, MO, USA) or Fisher Scientific (St. Louis, MO, USA) and were chemical-grade or higher. 

### 2.2. Protein Preparation

IpaD, IpaB, and L-DBF were made as previously described [8,9,24]. When IpaB was present, the genes were co-expressed with the chaperone IpgC to maintain IpaB or L-DBF solubility and stability. Briefly, *E. coli* Tuner (DE3) strains containing the genes in a pET expression plasmid were grown in LB media, with IPTG (1 mM) added to induce expression. After a 3 h induction, the bacteria were collected by centrifugation, resuspended in Binding Buffer, and lysed by sonication. The solution was clarified, and the supernatant loaded onto an IMAC column (immobilized metal affinity column; Cytiva, Marlborough, MA, USA). The proteins were eluted with a gradient of elution buffer containing imidazole. The fractions were collected and passed through a Q column to remove LPS. The IpaD was dialyzed against PBS and frozen at −80 °C. To remove the IpgC from the IpaB and L-DBF, LDAO (lauryl-dimethylamine oxide) was added to 0.05% and the sample passed over another IMAC column to remove the His6-tagged IpgC. The flow-through was collected and concentrated. It was dialyzed against 60 mM histidine pH 6.8, 150 mM NaCl, 5% sucrose, and 0.05% LDAO and stored at −80 °C. LPS levels were determined using a NexGen PTS with EndoSafe cartridges (Charles River Laboratories, Wilmington, MA, UDS). All proteins had LPS levels < 5 Endotoxin units/mg protein.

### 2.3. Preparation of L-DBF BECC438 or BECC470/ME Formulations

Squalene (8% by weight) and polysorbate 80 (2% by weight) were mixed to achieve a homogenous oil phase. Using a Silverson L5M-A standard high-speed mixer, 40 mM histidine (pH 6) and 20% sucrose were added to the oil phase and mixed at 7500 RPM, followed by six passes in a Microfluidics 110P microfluidizer at 20,000 psi to generate a milky emulsion of 4XME (MedImmune Emulsion) [25]. BECC470 (2 mg/mL) was prepared in 0.5% triethylamine by vortexing, followed by sonicating for 30 min in a 60 °C water bath sonicator until the BECC470 was completely dissolved. The pH of the BECC470 solution was adjusted to 7.2 with 1 M HCl. To make the L-DBF with the ME and BECC470 formulation, ME and BECC470 were mixed by vortexing for 2 min and incubated overnight at 4 °C. The next day, L-DBF was mixed with the ME-BECC470 solution at a volumetric ratio of 1:1 to achieve the desired final antigen concentration. The formulation processes for BECC438 were similar to the procedures described for BECC470 formulation [10,26].

### 2.4. Mice and Immunizations

Approval for the animal protocols was obtained from the Institutional Animal Care and Use Committees of the University of Kansas (Protocol: AUS 222-01). The young mouse experiments specifically conducted for bulk RNA sequencing were approved in the animal protocols at the University of Missouri (Protocol: 38241).

A workflow for the animal study is presented in Figure 1. The study involved multiple stages, including immunization, pre-challenge assessments, survival tests, and RNA sequencing. Figure 1 outlines the sequence of immunizations and sample collections, highlighting the different age groups and experimental conditions. 

#### 2.4.1. Young Mice

Six-to-eight-week-old female BALB/c mice (n = 14/group; 10 for survival test, 4 for pre-challenge immune response assessment) (Charles River Laboratories, Wilmington, MA, USA) were used for young mouse experiments. Prior to administration, PBS or 2 μg L-DBF in BECC470-ME (L-DBF + BECC470/ME) or 2 μg L-DBF in BECC438-ME (L-DBF + BECC438/ME) was prepared in 30 µL volumes. For immunizations, mice were anesthetized using isoflurane and vaccine formulations administered intranasally (IN) as previously described. Immunizations were on days 0, 14, and 28 for this study. 

#### 2.4.2. Elderly Mice

Female mice older than eighteen months were used as the elderly mouse cohort and for bulk RNA sequencing. The sample size was n = 20/group (10 for survival test, 4 for pre-challenge immune response assessment, 3 for pre-challenge sample collection for bulk RNA sequencing, and 3 for post-challenge sample collection for bulk RNA sequencing). For bulk RNA sequencing, an additional set of six-to-eight-week-old female BALB/c mice (n = 15/group; 9 for survival test, 3 for pre-challenge immune response assessment, and 3 for post-challenge immune response assessment) were used to match the vaccine dose. Vaccine preparations, either PBS or 1 μg L-DBF in BECC470-ME (L-DBF + BECC470/ME) or 1 μg L-DBF in BECC438-ME (L-DBF + BECC438/ME), were prepared in 30 µL volumes. Mice were anesthetized with isoflurane and received immunizations via the intranasal (IN) route on days 0, 14, and 28. The dose of vaccine antigen was reduced for elderly mice due to their unstable health condition at time of vaccination. 

#### 2.4.3. Mature Mice

Female eight-to-eleven-month-old mice (n = 14/group; 10 for survival test, 4 for pre-challenge immune response assessment) (Charles River Laboratories, Wilmington, MA, USA) were used for mature mouse experiments. Mice were immunized with PBS or 2 μg L-DBF in BECC470-ME (L-DBF + BECC470/ME), following procedures that were similar to those in young mice. 

### 2.5. IgG and IgA ELISAs 

Fecal pellets and 100 μL blood obtained by the orbital sinus route were collected on days 27, 41, and 55 (Figure 1). All samples were collected exclusively from mice immunized with either PBS, 2 μg L-DBF/BECC470-ME (young and mature), or 1 μg L-DBF/BECC470-ME (elderly). The mouse fecal sample processing involves taking a balance with labeled 1.5 mL EP tubes, weighing fecal pellets, adding PBS with 0.2% sodium azide, vortexing at 2500 rpm for 40 min, centrifuging twice at 10,000–13,000 RPM (24 °C), transferring supernatant to tubes with PMSF, and storing samples at −20 °C until using. Anti-IpaD and -IpaB IgG and IgA titers were determined as previously described [10]. Briefly, microtiter plate wells were coated with 100 ng IpaB or IpaD in 100 µL PBS and incubated at 37 °C for 3 h. The wells were the blocked with 10% nonfat dry milk in PBS overnight. Sera were added to the wells in duplicates as the primary antibody for a 2 h incubation at 37 °C. After washing with PBS-0.05%, Tween, an HRP–secondary antibody (IgG(H + L), 1:1000; IgA, 1:500), was added and incubated for 1 h at 37 °C. After an additional wash, OPD substrate (o-phenylenediamine dihydrochloride) was added and detected at 490 nm by ELISA plate reader. Endpoint titers were determined by fitting antibody titrations to a five-parameter logistic model. 

### 2.6. Shigella Challenge Studies

*Shigella* challenge strains were streaked onto tryptic soy agar containing 0.025% Congo red and incubated at 37 °C overnight and subcultured in tryptic soy broth (TSB) at 37 °C, until A600 reached 1.0. Bacteria were harvested by centrifugation, resuspended in PBS, and diluted to 1 × 10^6^ CFU in a 30 µL volume for IN challenge. Mice were monitored twice a day for weight loss and health scores for two weeks. Mice were euthanized if their weight loss exceeded 25% of their original weight for more than 72 h or their blood glucose reached ≤100 mg/dL with poor health scores. All remaining mice were euthanized on day 14 post-challenge (Figure 1) [9].

### 2.7. Cytokine Determinations

Lung cells were collected on day 53 (three days before a challenge of the remaining mice) and incubated with 10 µg/mL IpaB, IpaD or PBS for 48 h at 37 °C. Supernatants were collected and analyzed with U-PLEX kits for cytokines IFN-γ, IL-17A, IL6, and TNF-α. Cytokine concentrations were determined using an MSD plate reader with associated analytical software version 4.0 (Meso Scale Discovery, Rockville, MD, USA). All samples were collected solely from mice immunized with either PBS, 2 μg L-DBF/BECC470-ME (young and mature) or 1 μg L-DBF/BECC470-ME (elderly).

### 2.8. Bulk RNA Sequencing 

Total RNA was isolated from the lung cells that were collected on day 53 (three days before a challenge of the remaining mice) or collected on day 59 (three days after the challenge) using the RNeasy^®^ Mini kit according to the manufacturer’s instructions (QIAGEN, Hilden, Germany) (Figure 1). All samples were obtained solely from mice immunized with either PBS or 1 μg L-DBF/BECC470-ME (young and elderly). RNAs with an RNA Integrity Number (RIN) > 7 were shipped to CD Genomics (Shirley, NY, USA) for bulk RNA sequencing. The RNA-seq read count data were then processed for differential expression and pathway analysis.

### 2.9. RNA Seq Data Preprocessing and Normalization

The raw sequencing data were first preprocessed to remove low-quality reads and adaptors using fastp (v0.02.0) [27], then quality control was performed using FastQC (v0.11.9) (https://www.bioinformatics.babraham.ac.uk/projects/fastqc/ (accessed on 10 April 2023)). The clean reads were then aligned to the mouse reference genome (GRCm39-mm39) using STAR (2.7.10) with default parameters [28]. Gene expression levels were quantified, and the resulting count data were normalized using the DESeq2 package (v1.42.0) [29] in R (v4.0) [30]. The DESeq2 package implements a method that normalizes differences in library size by estimating size factors [29]. Given a matrix of read counts, the size factor for each sample is calculated, such that the median ratio of a gene’s read count in that sample to the geometric mean of that gene’s read count across all samples is close to 1. The gene-wise dispersion estimate and mean-dispersion relationship estimate were assessed, and the final dispersion values were obtained by combining the gene-wise dispersion estimates with the fitted mean-dispersion relationship.

### 2.10. Differential Expression Analysis

The data were fitted to a negative binomial distribution, and the Wald test was used to identify differentially expressed genes (DEGs) between different experimental groups, such as vaccinated mice vs. naïve mice (V vs. N), naïve infected mice vs. naïve mice (NI vs. N), vaccinated infected mice vs. vaccinated mice (VI vs. V), and vaccinated infected mice vs. naïve infected mice (VI vs. NI) (Figure 2). DEGs were considered significant based on a *p*-value threshold 0.05 and log2 fold change of 1.5 and above.

### 2.11. Heatmaps

Heatmaps were generated to visualize the expression patterns of the DEGs across different samples. Hierarchical clustering was used to group genes with similar expression patterns. The heatmap was generated using the pheatmap package (v1.0.12) [31] in R. The RNA-seq count data were transformed using the regularized log transformation (rlog) from the DESeq2 package. This transformation is used to stabilize the variance across the range of counts to visualize the differentially expressed genes.

### 2.12. Volcano Plots

Volcano plots were used to visualize the relationship between fold change and the statistical significance of the DEGs. The Enhanced Volcano package (v1.20.0) [32] in R was employed to create the plot, highlighting significant DEGs with a log2 fold change above 1.5 and below −1.5 and an adjusted *p*-value (0.05).

### 2.13. Gene Ontology (GO) Enrichment Analysis

GO enrichment analysis finds significantly overrepresented or underrepresented terms of biological processes or functions associated with a gene set of interest. The GO enrichment analysis was performed by calculating the expected and observed frequencies of GO terms in the gene list and then applying Fisher’s exact test to determine if the observed frequency is significantly higher than that expected by chance. The number of genes associated with each GO term are summed, then these counts are plotted as bar graphs or dot plots. A network was created where the connections between nodes (edges) are based on shared genes using the clusterProfiler (v4.0) R package [33].

### 2.14. Principal Component Analysis (PCA)

Principal component analysis (PCA) was conducted to examine the overall variation in the data and to visualize the separation between different experimental groups. The PCA plot was generated using the plot PCA function in R to reduce the dimension of the dataset, and then the first two PCAs were plotted on the x and y axes. The top 500 most variable genes for the PCA plot were used [34].

### 2.15. Immune Pathway and Gene Cluster Analyses

The immunoglobulin production, the production of molecular mediators of immune response, and T cell differentiation in both young and elderly mice were investigated for changes across all infection comparisons (NI vs. N, VI vs. V, VI vs. NI). Counts of genes associated with each GO term were summed and plotted as bar graphs. The GO term pathways were then categorized into B cell response-related, T cell response-related, and innate response-related categories by manual curation, evaluating their changes in total counts across all infection comparisons (NI vs. N, VI vs. V, VI vs. NI) in both age groups. The total counts of genes associated with each category were summed and plotted as violin graphs. Genes were classified into various immune cell subsets and MHC II-related genes by manual curation. The total log2 fold change of genes from the DEGs data associated with each category was summed and plotted as stacked bar graphs. Positive counts or a log2 fold change indicated upregulation, while negative values represented downregulation. The Immgen database (http://rstats.immgen.org/MyGeneSet_New/index.html (accessed on 10 February 2024)) was used to support the manual curation. 

### 2.16. Statistical Analyses

GraphPad Prism 8.1.2 was used to prepare data and perform statistical analyses. PBS groups were compared with the other vaccinated groups using Dunnett’s multiple comparison test. A *p* value of <0.05 was considered significant (* *p* < 0.05, ** *p* < 0.01, *** *p* < 0.001).

## 3. Results

### 3.1. Different Ages of Mice Immunized with Multimeric LDBF + BECC470 Formulated in an Oil-in-Water Emulsion Experience Protection against Shigella Infection

BECC470 is a bacterial lipid A analogue that has been found to enhance the mouse immune response to immunization [22]. Additionally, a VLP-based vaccine adjuvanted with BECC470 was shown to induce protective mucosal and systemic responses against SARS-CoV-2 in mice [23]. Our previous studies showed that L-DBF in an oil-in-water emulsion called ME (L-DBF/ME), admixed with a different BECC, BECC438, could induce effective protection against *Shigella* spp. infections in young mice [10]. However, prior studies involving viral pathogens and other unpublished work suggested that BECC470 might be more effective than BECC438 against intracellular pathogens. Therefore, we chose to assess the immune response elicited by 2 µg L-DBF/ME + 1 µg BECC470 or BECC438 (2 µg L-DBF/ME/BECC470 or BECC438) against lethal *Shigella* spp. infections in young mice (6–8 weeks; n = 10) and by 1 µg L-DBF/ME/BECC470 or BECC438 in elderly mice (>18 months; n = 10) (Figure 1). This dose modification for the elderly mice was mandated by the Animal Care Unit because the inclusion of multiple adjuvants produced unexpected adverse reactions, including weight loss and distress, after they had been housed for two years. This adjustment was necessary to ensure the health and well-being of the animals, while optimizing the immune response. All young mice that were vaccinated with 2 µg L-DBF/ME/BECC470 or BECC438 survived the challenge (1 × 10^6^ CFU/mouse), while 90% of the elderly mice vaccinated with 1 µg L-DBF/ME/BECC470 survived the challenge. Notedly, only 60% of the elderly mice vaccinated with 1 µg L-DBF/ME/BECC438 survived the challenge. All mice in the PBS group succumbed to the challenge in essentially the same window of time (Appendix A). The vaccinated young mice started regaining weight after two days following the challenge, with the elderly group only doing so after three days (Figure 3 and Appendix A). Given the superior performance of L-DBF/ME/BECC470 in elderly mice compared to L-DBF/ME/BECC438, subsequent studies focused on the immune responses elicited by L-DBF/ME/BECC470 across various age groups.

We subsequently evaluated the immune responses of vaccinated mice before the challenge, identifying age-dependent variations in antibodies and cytokine secretion induced by the L-DBF/ME/BECC470 formulation. Increases in anti-IpaD and -IpaB IgG and IgA were observed in response to vaccination in both age groups (Appendix A). In young mice, an elevated secretion of IFN-γ and IL-17 in the lungs was observed (Figure 3A). This cytokine profile indicates potent Th1 and Th17 immune responses, which are indicative of effective cellular immunity and controlled inflammation [10,35]. In elderly mice, IL-17 secretion remained elevated, but IFN-γ secretion was suppressed relative to young mice, presenting a distinct Th17 dominance (Figure 3B). Notably, IL-17 secretion in elderly mouse lungs reached nearly twice the levels observed in other age groups. These findings collectively highlight age-associated variations in the immune response induced by L-DBF/ME/BECC470 following subsequent challenge. The heightened IL-17 secretion in elderly mice, coupled with suppressed IFN-γ levels, suggests a shift towards a Th17-skewed response in this age group (Figure 3). 

In elderly mouse lung samples, we found a significant rise in IL-6 levels when stimulated with media (Appendix A). However, there were no statistically significant differences for those cytokines in the young mouse lung samples. In spleen samples from the vaccinated group, our findings revealed significant immune responses in young mice, with the levels of IFN-γ, IL-17, and IL-6 stimulated with IpaB or IpaD being all significantly higher than the negative controls (Appendix A). Interestingly, we noted a unique observation in elderly mice: the IL-6 levels for cells stimulated with IpaB in the negative controls were found to be higher than those in the vaccinated group (Appendix A). While this altered cytokine milieu did not compromise the overall vaccine efficacy, it supports the potential necessity of tailoring vaccine formulations to distinct age populations. 

We also assessed the efficacy of the L-DBF/ME/BECC470 formulation in mature mice aged 8–11 weeks (Appendix A). All mature mice vaccinated with 2 µg of L-DBF/ME/BECC470 survived the challenge of 1 × 10^6^ CFU/mouse, whereas all mice in the PBS group succumbed to the challenge (Appendix A). Following the challenge, vaccinated mature mice began to regain weight three days later, a trend resembling that observed in the elderly group. However, the degree of weight loss experienced was less pronounced (Appendix A) compared to the elderly group (Appendix A). Increases in anti-IpaD and -IpaB IgG and IgA were observed following vaccination in mature groups (Appendix A), showing levels closely resembling those observed in the young group (Appendix A). A heightened secretion of IFN-γ and IL-17 was detected in the lung cell suspensions of vaccinated mature mice (Appendix A), a trend mirroring that observed in the young mice (Figure 3A). Notably, IL-6 levels in lung samples increased upon media stimulation, and TNF-α levels rose when subjected to IpaB stimulation (Appendix A). However, in mature mice, only IL-6 levels in cultured splenocytes were significantly elevated after IpaD stimulation, compared with the levels in the control cultures (Appendix A). 

As young and elderly populations are inordinately affected by shigellosis [36], our subsequent studies primarily targeted these age groups. To standardize the vaccine dosage between those two age groups, an additional trial was performed, wherein young mice were vaccinated with 1 µg of L-DBF/ME/BECC470 for a comparative analysis (n = 10). This trial demonstrated 100% protection following the same challenge dose, whereas all mice in the PBS group succumbed to the challenge (Appendix A).

### 3.2. Differential Gene Expression in Vaccinated Young or Elderly Mice before Infection

To evaluate the baseline immune response and the direct impact of the vaccine on the host immune system and to shed light on age-related differences in the response to vaccination, we first focused on the transcriptomic changes in the lungs of the young and elderly mice following vaccination with three doses of 1 µg L-DBF/ME/BECC470 (vaccinated vs. naïve). Bulk mRNA sequencing was conducted to compare the gene expression profiles on day 3 before infection (day 53 of total workflow; V vs. N, Figure 1 and Figure 2). Although cytokine levels were measured at different doses, our bulk RNA sequencing analysis, performed with the same dose across all age groups, provided a valid basis for this comparison. In young mice, the transcriptomic analysis highlighted a robust and diverse response to vaccination. A total of 248 genes were significantly upregulated, indicating the activation of multiple immune pathways (*p* < 0.05; Figure 4A and Appendix A). Notably, many of these upregulated genes were associated with B cell responses, including genes related to immunoglobulins (heavy constant: *Ighg2b*, *Ighg2c*; kappa light chains: *Igkv14-126*, *Igkv16-104*, *Igkv3-10*, *Igkv3-5*, *Igkv3-7*, *Igkv4-50*, *Igkv4-63*, *Igkv5-48*, *Igkv8-21, Igkv8-24*), Fc or immunoglobulin receptors (*Fcrlb*, *Pigr*), inflammatory response (*Il17a*, *Il17f*, *Mmp12*), chemokines and chemokine receptors (*Ccl8*, *Ccr6, Cxcl13*, *Cxcl9*), cell signaling and communication (*Gcsam*, *H2-DMb1*), and cell adhesion and migration (*Cxcr6*, *Slamf8*) (log2FoldChange > 2; Figure 4C). We also observed an upregulation of genes related to MHC-II (*H2-Aa*, *H2-Ab1*, *H2-Ea*, *H2-Eb1*; 1.3 < log2FoldChange < 2, Appendix A). The upregulation of these genes suggested an orchestrated immune and cellular response, potentially involving processes such as inflammation, immune cell recruitment, and antigen presentation. Conversely, 152 genes were significantly downregulated in young mice in response to vaccination (*p* < 0.05; Figure 4A and Appendix A). Many of these downregulated genes were associated with transport and metabolism (*Slco4c1*, *Cyp4a32*, *Cyp1a1*, *Slc34a2*, *Slc16a9*) (log2FoldChange < −1.3; Appendix A). We found that four genes—*Chia1*, *Mbl1*, *Icam5*, and *Kcne2*—exhibited marked downregulation (log2FoldChange < −2), which relate to cell adhesion, immune responses, inflammation, and channel function (Figure 4C). 

In contrast, elderly mice showed fewer alterations in gene expression changes following vaccination, with 12 genes upregulated and 2 genes downregulated (*p* > 0.05; Figure 4B). Nevertheless, these upregulated genes were mainly related to chemokines and immunity (*Ccl8*, *Il17a*, *Il17f*, *Il17re*, *Pigr*, log2FoldChange > 2; Figure 4D), suggesting that elderly mice did initiate an immune response to the vaccine. Moreover, one gene—*Iglv2*, an immunoglobulin light-chain variable region—exhibited significant downregulation (log2FoldChange < −2; Figure 4D). This suggests that the elderly mice may not mount a humoral response as substantial as young mice. This downregulation implies that the immune response in the vaccinated mice at a young age is different from that of older mice, possibly to prevent excessive inflammation and modulate immune functions. 

We found *Ccl8* and *Pigr* were upregulated in both young and elderly mice following L-DBF/ME/BECC470 vaccination. *Ccl8* is a key gene involved in recruiting immune cells to sites of infection [37], and *Pigr* is responsible for transporting immunoglobulins across epithelial barriers to defend against pathogens [38]. This shared response in both young and elderly mice highlights the capacity of L-DBF/ME/BECC470 to stimulate the mobilization of immune cells. Moreover, the upregulation of the Th17-related genes *Il17a* and *Il17f* was also seen for both young and elderly vaccinated mice, while the upregulation of Il17re was only found in vaccinated elderly mice. *Il17re* codes for IL-17 receptor E, a member of the IL-17 receptor family primarily found on epithelial cells, keratinocytes, and Th17 cells [39]. Studies have shown that *Il17re*-transgenic Th17 cells produce higher levels of IL-17. The upregulation of *Il17re* in vaccinated elderly mice may contribute to the elevated IL-17 secretion observed in this age group (Figure 3). 

### 3.3. Gene Expression Profiling between Naïve Infected Mice vs. Naïve Mice in Different Age Sets

To understand the dynamics of the host response and to identify potential age-related factors affecting susceptibility to *Shigella* infection, we conducted a comparative analysis of transcriptome differences between naïve infected (NI) mice on day 3 post-infection (day 59 of total workflow) and naïve (N) mice (day 53 of total workflow) across both the young and elderly age groups (NI vs. N; Figure 2). A transcriptome analysis of young NI vs. N mice showed significant differences in gene expression. A substantial number of genes (4485) were upregulated (*p* < 0.05; Appendix A), including those related to immune response (*Fcrlb*, *Fcgr2b*, *Pigr*, *C3ar1*, *Ly6i*, *Ifitm6*, etc.), chemokines and cell migration (*Cxcr2*, *Cxcl3*, *Cxcl9*, *Cxcl13*, *Ccl2*, *Ccl7*), and antimicrobial and defense responses (*S100a8*, *S100a9*, *Mmp9*, *Lcn2*, *Prok2*) (log2FoldChange > 1.3; Appendix A). Interestingly, we found *Il17a* significantly upregulated in NI mice; however, the *Il17f* expression was not. The top ten upregulated pathways (Appendix A) were involved in the following: positive regulation of the defense response, leukocyte migration, leukocyte chemotaxis, regulation of response to biotic stimulus, cell chemotaxis, myeloid leukocyte migration, regulation of innate immune response, defense response to bacterium, activation of immune response, and positive regulation of response to biotic stimulus. Conversely, 4153 genes were downregulated in young mice after infection (*p* < 0.05; Appendix A). Most of these genes are related to T cell receptors (*Traj11*, *Traj12*, *Trbj1-5*, *Trbj2-2*, *Trbv12-1*, *Trbv12-2*, *Trdv1*, *Trdv2-1*, *Trgv7*, etc.) and immune regulation (*Acan*, *Aire*, *Ccr9*, *Cd8a*, *Cd8b1*, etc.) (log2FoldChange < −1.3; Appendix A). *Il17rb* was found to be significantly downregulated in NI mice. The top ten downregulated pathways (Appendix A) were primarily associated with adaptive immune responses, including the following: the T cell receptor signaling pathway, T cell differentiation, gamma–delta T cell activation, gamma–delta T cell differentiation, positive regulation of lymphocyte activation, immunoglobulin production, positive regulation of T cell activation, intermediate filament organization, positive regulation of cell activation, and positive regulation of leukocyte activation. 

In contrast to young mice, there were fewer changes in gene expression in elderly mice post-infection (day 59 of total workflow). Among the differentially expressed genes, 1011 were upregulated and 875 were downregulated (*p* < 0.05; Appendix A) in elderly NI mice. Among the upregulated genes, those associated with immune response (*C3ar1*, *Ccl2*, *Csf3*, *Fcrlb*, *Ifitm6*, *S100a8*, etc.) and metabolism (*Awat1*, *Cpa6*, *Gdpd2*, *Hkdc1*, etc.) were also upregulated in young NI mice (log2FoldChange > 1.3; Appendix A). Other upregulated genes in elderly NI mice are not as prominent in young NI mice, including those associated with tissue development and maintenance, such as *Ankrd22*, *Cstdc4*, *Cstdc5*, *Cstdc6*, and *Il1r2*. The top ten upregulated pathways (Appendix A) in elderly mice were mainly associated with: skin development, wound healing, acute inflammatory responses, regulation of body fluid levels, organic anion transport, keratinocyte differentiation, blood coagulation, carboxylic acid transport, organic acid transport, and hemostasis. In contrast to young NI mice, which exhibited a robust immune response to combat infection, elderly mice prioritize mechanisms aimed at tissue repair and inflammation control. Notably, while young NI mice mainly downregulated T cell receptor-related genes, elderly NI mice downregulated both T cell-related genes (*Cd3g*, *Cd4*, *Cd5*, *Cd7*, *Cd8a*, *Cd8b1*, *Lck*, *Rag1*, *Rag2*, *Tcf7*, *Themis*, *Trat1*, *Trbv*, *Trdv1*, *Trgc2*, *Trgv1*), and B cell-related genes (*Igkv4-74*, *Igkv7-33*, *Iglv2*) (log2FoldChange < −1.3; Appendix A), indicating a further suppression of immune responses during *Shigella* infection with aging. The top ten downregulated pathways (Appendix A) in elderly mice included the following: immunoglobulin production, production of molecular mediators of immune response, nuclear chromosome segregation, chromosome segregation, mitotic sister chromatid segregation, sister chromatid segregation, lymphocyte differentiation, leukocyte mediated immunity, lymphocyte mediated immunity, and T cell differentiation. Notably, the downregulation of immunoglobulin production and T cell differentiation genes in elderly naïve infected (elderly NI) mice was greater than those in young naïve infected (young NI) mice, which indicated potential age-related alterations in immune responses, highlighting a diminished immune function in elderly mice compared to young mice. These findings served as a control to evaluate the natural immune response to infection, thereby providing a foundational understanding that aided in elucidating how L-DBF/ME/BECC470 induces protective responses during *Shigella* infection in both young and elderly age groups.

### 3.4. Gene Expression Profiling between Vaccinated Infected Mice vs. Vaccinated Mice in Different Age Sets

To identify differentially regulated genes and pathways indicative of a vaccine-induced protective response during *Shigella* infection, we then analyzed gene expression between vaccinated infected (VI) mice (day 59 of total workflow) and vaccinated (V) mice (day 53 of total workflow; VI vs. V, Figure 2). In young VI mice, 1148 genes were upregulated and 1171 genes were downregulated, relative to their uninfected counterparts (*p* < 0.05; Appendix A). The upregulated genes in young VI mice encompassed a diverse range of functions related to immune response and inflammation, such as pro-inflammatory cytokines (*Il17a*, *Il17f*, *Il1rn*, *Il36g*), chemokines and receptors (*Cxcl3*, *Cxcl9*, *Cxcl10*, *Cxcl13*, *Ccl8*), antimicrobial response (*Nlrp12*, *Arg1*), and immune cell regulation (*Itgam*, *Ceacam16*, *Cd177*, *Ifitm1*, etc.) (log2FoldChange > 1.3; Appendix A). Notably, both *Il17a* and *Il17f* were significantly upregulated in young VI. Upregulated genes were associated with processes encompassing the negative regulation of peptidase activity, negative regulation of hydrolase activity, negative regulation of proteolysis, regulation of peptidase activity, humoral immune response, antimicrobial humoral response, regulation of neurotransmitter levels, amide transport, negative regulation of endopeptidase activity, and amine transport (Appendix A). We found that several immune response genes were downregulated, such as genes related to T cell receptors and co-receptors (*Cd8a*, *Cd8b1*, *Ptcra*, *Rag1*, *Rag2*) (log2FoldChange < −1.3; Appendix A). The downregulated genes represented a regulatory response to prevent excessive inflammation or modulate specific pathways (Appendix A), such as the following: sperm motility, flagellated sperm motility, cilium movement involved in cell motility, cilium- or flagellum-dependent cell motility, cilium-dependent cell motility, keratinization, V(D)J recombination, intermediate filament organization, regulation of flagellated sperm motility, and cilium movement. 

In the transcriptomes of elderly VI mice, 1146 genes were upregulated, while 617 genes were downregulated relative to elderly V mice (*p* < 0.05; Appendix A). The upregulated genes in elderly VI mice were related to immune response (*Cd177*, *Clec4d*, *Clec4e*, *Csf3*, *Csf3r*, *Il1a*, *Il1bos*, *Il1r2*, *Il1rn*, *Il23a*, *Il36g*, *Ifitm1*, *Ifitm6*, *Trem1*, *Trem3*, etc.), chemokines and receptors (*Cxcl1*, *Cxcl2*, *Cxcl3*, *Cxcl5*, *Cxcr2*), and immune cell regulation (*Cd14*, *Cd33*, *Ceacam10*, *Fcrlb*) (log2FoldChange > 1.3; Appendix A). Upregulated processes (Appendix A) encompassed the following: myeloid leukocyte migration, granulocyte migration, leukocyte chemotaxis, neutrophil migration, leukocyte migration, granulocyte chemotaxis, neutrophil chemotaxis, acute inflammatory response, cell chemotaxis, and a defense response to bacterium. These findings suggest that elderly mice mount an immune response that prioritizes the recruitment and activation of immune cells, particularly myeloid and granulocytic cells, in response to the infection. Conversely, we found a cluster of downregulated genes related to immunoglobulin (*Igha*, *Ighv5-9*, *Igkv2-109*, *Igkv3-12*, *Igkv3-5*, *Igkv3-7*, *Igkv4-57*, *Igkv4-58*, *Iglv3*, etc.), genes associated with immune cell regulation and responses (*Cd200r3*, *Cd209a*, *Cd209d*, *Fcrl2*, etc.), and chemokine receptors (*Ccr3*, *Ccr9*) (log2FoldChange < −1.3; Appendix A). Downregulated pathways included the following: immunoglobulin production, production of molecular mediators of immune response, xenobiotic metabolic process, steroid metabolic process, cellular response to xenobiotic stimulus, olefinic compound metabolic process, the epoxygenase P450 pathway, fatty acid metabolic process, response to xenobiotic stimulus, and arachidonic acid metabolic processes (Appendix A). 

The outcomes of VI vs. V suggested that, despite a similar vaccine efficacy (100% in young vs. 90% in elderly mice; Appendix A), the vaccine-induced immune response post-infection differed between the young and elderly age groups. Both age sets exhibited a more focused immune response (Appendix A) compared to the NI vs. N comparison (Appendix A), with fewer regulated genes found. In young mice, vaccine-induced protective responses targeted regulating immune function and preventing excessive inflammation, while in elderly vaccinated mice, there was an activation of myeloid and granulocytic cells compared to the NI vs. N comparison. However, a downregulation of B cell responses (immunoglobulin production) was still observed in the elderly mice, indicating a potential age-related suppression of adaptive immune responses.

### 3.5. Age-Dependent Variations in Gene Expression and Pathway Alterations Related to Vaccine-Induced Immune Responses after Infection

To understand how L-DBF/ME/BECC470 altered the host response to lethal *Shigella* spp. infection, we compared the gene expression between vaccinated infected (VI) mice and naïve infected (NI) mice (Figure 2). A total of 2613 genes were upregulated, and 3301 genes were downregulated in young VI mice compared to young NI mice (*p* < 0.05; Figure 5A). The upregulated genes in young VI mice exhibited strong enhancements in immune response compared to the NI mice, such as T cell activation (*Il17a*, *Il17f*, *Il21*, *Rag1*, *Rorc*, etc.) and B cell functions (*Ighg2b*, *Ighg2c*, *Ighv1-33*, *Ighv14-2*, *Igkv11-125*, *Igkv12-44*, etc.) (log2FoldChange > 1.3; Appendix A). The upregulated pathways (Figure 5C) included the following: immunoglobulin production; production of molecular mediators of immune response; an antigen receptor-mediated signaling pathway; lymphocyte differentiation; positive regulation of T cell activation; positive regulation of leukocyte cell–cell adhesion; the T cell receptor signaling pathway; positive regulation of lymphocyte activation; positive regulation of leukocyte activation; and T cell differentiation. Conversely, our analysis revealed the downregulation of a variety of genes involved in different cellular processes, such as immune response and inflammation (*C3ar1*, *Camp*, *Ccl2*, *Cd177*, *Csf3*, *Itgam*, etc.), cell signaling and receptors (*Cxcr2*, *Fosl1*, *G0s2*, *Ggt1*, *Htr2c*, *Ifitm1*, *Ifitm6*, etc.), and cell adhesion and the extracellular matrix (*Ceacam10*, *Ceacam18*, *Chia1*, *Chst13*, *Csta3*, *Cstdc4*, *Cstdc5*, etc.) (log2FoldChange < −1.3; Appendix A). The top downregulated pathways found in young VI mice, compared to young NI mice, were as follows (Figure 5E): extracellular structure organization; a defense response to bacterium; extracellular matrix organization; external encapsulating structure organization; myeloid leukocyte migration; cell chemotaxis; leukocyte chemotaxis; negative regulation of peptidase activity; granulocyte migration; and peptide cross-linking. 

The same comparison in elderly mice (VI vs. NI) identified a total of 354 upregulated genes and 404 downregulated genes (*p* < 0.05; Figure 5B). Although the gene expression in the comparison of VI vs. NI in elderly mice exhibited fewer differences than those in young mice, we found genes in elderly VI mice upregulated compared to elderly NI mice, involving genes related to immune response and inflammation (*Il17a*, *Il17f*, *Il21*, *Il23r*, *Il27ra*, *Tnfsf11*, *Tnfsf8*, *Prf1*, *Ccl21a*, *Ccl8*, *Ccr4*, *Gzma*, *Gzmb*, *Gzmk*, *Gbp8*, *Gpr183*, *Tlr12*, *Nkg7*, *Cd3g*), T cell activation and signaling (*Cd28*, *Cd3d*, *Cd4*, *Cd40lg*, *Icos*, *Itgae*, *Lat*, *Lck*, *Rasgrp1*, *Rasal3*, *Zap70*), B cell activation and signaling (*Blnk*, *Cd19*, *Pax5*, *Pdcd1*), regulatory T cell markers (*Ctla4*, *Foxp3*, *Ikzf3*), and cytokines and chemokines (*Il12rb1*, *Ccl21a*, *Ccl8*, *Ccr4*, *Ccr6*, *Gpr183*) (log2FoldChange > 1.3; Appendix A). The top ten upregulated pathways (Figure 5D) included the following: immunoglobulin production; production of molecular mediators of immune response; natural killer cell-mediated immunity; lymphocyte-mediated immunity; leukocyte-mediated immunity; the regulation of natural killer cell-mediated immunity; natural killer cell-mediated cytotoxicity; antigen receptor-mediated signaling pathway; T cell receptor signaling pathway; and cell killing. The downregulated genes in elderly VI mice compared to NI mice covered a range of functions, such as signaling pathways and growth factor regulation (*Gdnf*, *Nppa*, *Nrtn*, *Tgfa*, *Thbs1*, *Wnt7b*, *Wnt9b*), cell adhesion and interaction (*Cldn3*, *Crb3*, *Marco*, *Vldlr*), and ion and amino acid transport (*Slc12a2*, *Slc12a6*, *Slc38a4*, *Slc6a12*) (log2FoldChange < −1.3; Appendix A). The top ten downregulated pathways (Figure 5F) included the following: keratinization; epidermis development; epidermal cell differentiation; skin development; keratinocyte differentiation; the neuropeptide signaling pathway; organic anion transport; intermediate filament organization; the negative regulation of peptidase activity; and carboxylic acid transport. 

Finally, we used principal component analysis (PCA) to assess the distinctions between VI and NI mice within the young and elderly age sets (Figure 6). Principal component analysis (PCA) revealed a clear separation between VI and NI young mice (Figure 6A). This segregation showed an impact of vaccination on the transcriptomic profile, which suggested a robust response in young mice following vaccination and subsequent *Shigella* infection. Unlike young mice, the PCA results for the elderly age set showed less separation between VI and NI mice, which suggested that the transcriptomic differences induced by vaccination in elderly mice might not be as distinct as those observed in the young age set (Figure 6B).

### 3.6. Comparative Analysis of B Cell, T Cell, and Innate Responses in Young and Elderly Mice following Vaccination and Infection

The noticeable upregulation of immunoglobulin production, the production of molecular mediators of immune response, and T cell differentiation in both young and elderly mice following vaccination and infection (VI vs. NI) prompted further investigation into the changes across all comparisons (V vs. N, NI vs. N, VI vs. V, VI vs. NI; Figure 7A). Before the infection, immunoglobulin production and the production of molecular mediators was upregulated in vaccinated mice compared to naïve mice in both young and elderly mice (V vs. N), with a reduced level of upregulation in the elderly mice compared to young mice. Following a *Shigella* infection, immunoglobulin production and T cell differentiation were consistently downregulated in the naïve–infected state (NI vs. N) in both ages, with a greater decrease observed in elderly mice. Conversely, immunoglobulin production was upregulated in young vaccinated mice post-infection (VI vs. V), while still exhibiting a downregulation in elderly mice, though less pronounced than in the naïve state (NI vs. N). Notably, the downregulation of T cell differentiation was not observed in either young or elderly vaccinated mice post-infection (VI vs. V). Interestingly, the production of molecular mediators of the immune response showed an upregulation in young naïve and vaccinated mice following infection (NI vs. N, VI vs. V), while it was downregulated in both naïve and vaccinated elderly mice after infection (NI vs. N, VI vs. V), indicating a differential response between the age groups. 

The observed differences in the regulation of those three pathways above, between young and elderly mice, following vaccination and infection suggested that the L-DBF/ME/BECC470 formulation employed distinct mechanisms to protect mice against *Shigella* infection in the two age groups. To further investigate this, we categorized the pathways into B cell response-related, T cell response-related, and innate response-related categories (Figure 7B) and evaluated their changes across all comparisons (V vs. N, NI vs. N, VI vs. V, VI vs. NI) in both age groups. Before the infection, vaccination resulted in an upregulation of B cell response-related pathways in young mice, suggesting a primed adaptive immune response. However, some innate response-related pathways were downregulated, possibly reflecting a reduced inflammatory state post-vaccination. In elderly mice, some B cell and T cell response-related pathways were upregulated, but the extent of the upregulation was less pronounced compared to young mice.

We then found that these three immune response pathways were upregulated in both young and elderly mice following vaccination and infection (VI vs. NI; Figure 7B). However, significant differences were found in the naïve state (NI vs. N; Figure 7B), where B cell, T cell, and innate response-related pathways were predominantly upregulated, with some instances of downregulation in young mice. Meanwhile, in elderly mice, these pathways were mainly downregulated. Interestingly, the regulatory patterns in the vaccinated state (VI vs. V; Figure 7B) were less significant in young mice, whereas a more pronounced upregulation and fewer downregulation events were evident in elderly VI vs. V mice. The alterations observed in the immune response pathways indicated that in young mice, vaccination primarily triggered a targeted B cell immune response, an augmented T cell response, and a mitigated inflammatory response, contributing to robust protection against *Shigella* infection. Conversely, in elderly mice, the L-DBF/ME/BECC470 mitigated the downregulation of B cell, T cell, and innate response pathways, potentially enhancing overall immune protection against *Shigella* infection. 

To delve deeper into the differences in immune responses between young and elderly mice following vaccination and infection, we categorized genes into various immune cell subsets and MHC II-related genes (Figure 8). We then examined these gene clusters across all comparisons (V vs. N, NI vs. N, VI vs. V, and VI vs. NI) in both young and elderly mice. Interestingly, both young and elderly mice exhibited a similar trend of upregulation in Th1, Th17, Treg, Tfh, B cells, dendritic cells (DC), and MHC II-related genes in the VI vs. NI comparison, with a slight downregulation observed in macrophages. Notably, in VI vs. NI, young mice displayed a slight downregulation of Th2, whereas all response-related genes were upregulated in this comparison in elderly mice. Furthermore, in the V vs. N comparison, young mice showed an upregulation of most immune responses, except for Treg, while elderly mice exhibited a downregulation of most responses, except for Th17 and macrophages. In the VI vs. V comparison, young mice displayed a more focused regulation of immune responses, with a slight downregulation observed in Th1 and a minimal upregulation of Th2. Conversely, most responses remained downregulated in elderly mice in the VI vs. V comparison, with fewer changes compared to the NI vs. N comparison. Notably, Tfh was upregulated in elderly mice in the VI vs. V comparison. These findings suggest age-related differences in immune responses to vaccination and infection and highlight the importance of considering age-related factors in the development and evaluation of vaccines.

## 4. Discussion

Shigellosis is a significant public health concern, particularly in developing countries with poor sanitation and hygiene practices [40]. Currently, global warming has led to an intensification of weather events, including storms and floods, in many parts of the world [41], and these events can create conditions conducive to the rapid spread of diarrheal infections, including shigellosis [41]. Vaccination offers a proactive solution to address changing climatic conditions and emerging antibiotic resistance. Investment in the research and development of a *Shigella* vaccine for vulnerable groups is a public health imperative [42]. This includes vaccine formulations that are safe and effective for the very young and old, who are at an elevated risk of morbidity and mortality. Because *Shigella* spp. are human-adapted pathogens, finding an ideal animal model for research poses a challenge [1,13]. Despite the emergence of a new mouse intestine model, acquiring enough aging mice and the associated costs present significant obstacles [12]. Therefore, the mouse lung model remains essential for early-stage *Shigella* vaccine development, providing a more accessible and cost-effective approach to studying the pathogen’s dynamics and thoroughly evaluating potential vaccine candidates. In previous studies, our lab explored formulations for *Shigella* spp. vaccines, focusing on T3SS subunit candidates with various adjuvants [9]. Combining low doses of self-adjuvanting L-DBF with a BECC438/ME formulation enhanced vaccine efficacy against *Shigella* spp. infection [10]. Because BECC438 did not provide effective protection in elderly individuals, we used BECC470 for this group and found that L-DBF/ME/BECC470 elicited protective immune responses across all age groups. Lung cells from elderly mice following vaccination displayed a distinctively elevated IL-17 secretion and suppressed IFN-γ levels after stimulation with IpaB or IpaD, which may suggest a unique Th17-skewed response in this age set. As noted, young mice showed a significant increase in IL-17 expression in the spleen after vaccination, whereas elderly mice did not, indicating that age affects the systemic immune response, particularly IL-17 production in the spleen [43]. Further research may be needed to understand the mechanisms behind the differences in cytokine responses in lung and spleen, and to optimize vaccination strategies for elderly populations.

A transcriptomic analysis of lung tissues following vaccination with L-DBF/ME/BECC470 revealed age-related differences in the immune response. In young V mice, a robust and coordinated immune response was observed, aimed at combating *Shigella* infection. This included the activation of long-lasting humoral immunity, effective antigen presentation, and predominant activation of Th1 and Th17 responses, specifically targeting the mucosal site of *Shigella* entry. These mechanisms likely contribute to preventing bacterial colonization and invasion in the young age group. In contrast, elderly V mice exhibited a lower immune response, potentially with a weaker humoral component compared to young mice. This supports existing studies showing that aging reduces B cell function through increased intrinsic B cell inflammation and reduced signal transduction [37]. Additionally, in young V mice, the upregulated expression of MHC II-associated genes implied an enhanced capacity for antigen presentation. This MHC II response would be critical for the activation of CD4+ T cells, leading to effective immune responses and vaccine-induced protection [44]. The greatly diminished upregulation in vaccinated elderly mice suggested there are differences in antigen presentation and T cell activation between the two age groups at day 28 following the last vaccination. Previous research has shown that MHC-II genes display a clear age-dependent relationship in both murine and human tissues [45,46]. Further research on the observed differences in MHC II gene expression may provide insights into the mechanisms underlying age-related variations in vaccine responses. Our findings underscore the challenges in eliciting strong B cell responses and antigen presentation in aging populations, highlighting the need for strategies to enhance this aspect of the immune response. Despite the diminished immune response in elderly mice, the upregulation of Th17-related genes and cytokines induced by L-DBF/ME/BECC470 still provided significant protection, with a 90% protection rate. This identifies the importance of Th17 responses in mucosal immunity and host defense against *Shigella* infection. Specifically, *Il17a* and *Il17f* were upregulated in both young and elderly vaccinated mice, while *Il17re* was specifically upregulated in elderly vaccinated mice. This upregulation in elderly mice may represent an effort to enhance the aging immune system or an altered immune response to adapt to vaccination. Further functional assays and pathway analyses will be instrumental in unraveling the significance of IL-17RE upregulation in elderly mice. Our previous work established the importance of Th17 responses in countering *Shigella* spp. infections using young mice [9,10]. Consistently, cytokine quantification revealed increased levels of IL-17 in the lungs of both young and elderly mice prior to challenge. By employing RNA-seq to investigate the Th17 response at the transcriptome level post-vaccination, we gained insights into the landscape of gene expression between different ages during this critical immune response. 

To gain a deeper understanding of how the L-DBF/ME/BECC470 formulation conferred protection against *Shigella* infection in different age groups, we compared gene expression patterns in both naïve and vaccinated mice across distinct age groups after *Shigella* challenge. Moreover, diverse innate and adaptive immune responses were elicited to combat the *Shigella* infection, even without prior vaccination in young mice (NI vs. N), while naïve elderly mice showed a suppression of immune-related genes and immune cell subsets, with upregulated pathways primarily related to tissue repair during infection. Such a decline in immunity could make elderly mice more susceptible to *Shigella* infection in the absence of vaccination. Conversely, the transcriptomic profile of young vaccinated infected (VI) mice revealed a focused immune response, with a regulatory response to maintain a balance to avoid acute inflammatory events during the infection (VI vs. V). We detected a significant upregulation of both *Il17a* and *Il17f* in young VI mice (VI vs. V), while only *Il17a* was found upregulated in young NI mice (NI vs. N), which suggested a more robust Th17 response following vaccination and subsequent infection. This enhanced Th17 response could contribute to more effective protection against *Shigella* spp. in the vaccinated group. Meanwhile, in elderly mice, a similar decrease in dysregulation was observed, but the overall immune response in elderly VI vs. V appeared to be less dysregulated compared to elderly NI vs. N. This modulation of immune response by L-DBF/ME/BECC470 in elderly mice led to an upregulation of immune responses in the comparison of elderly VI vs. NI. Notably, we observed an increased activation of myeloid and granulocytic cells in elderly VI vs. V, which indicated an aggressive innate immune reaction to *Shigella* infection in elderly vaccinated mice. These findings in elderly vaccinated mice are consistent with observations in human infections like SARS-CoV-2 [38,44]. 

Interestingly, the immune regulation was similar between young and elderly mice in the VI vs. NI comparison, as they both showed a protective immune response leading to effective pathogen clearance and resolution of infection. There was a slight downregulation of Th2 cells in young VI vs. NI, and a downregulation of macrophages in VI vs. NI at both age groups, indicating a focused and efficient immune response prioritizing Th1 and Th17 responses, crucial for combating intracellular pathogens like *Shigella* [45,46]. The differences seen in NI vs. N young and elderly mice indicate that young naïve mice could succumb to the overwhelming inflammatory response caused by *Shigella* infection, while elderly naïve mice might face increased mortality due to compromised immune defenses. However, the similarities between young and elderly mice in VI vs. NI suggest that in young mice post-vaccination, a heightened and well-coordinated immune response likely aids in better protection against *Shigella* infection by effectively combating *Shigella* and minimizing tissue damage. In elderly vaccinated mice, the vaccine still offers protection against *Shigella* infection by limiting the dysregulation of immunity, indicating that even a modest enhancement in immune function in elderly mice can be advantageous in fighting *Shigella*.

One finding in this study was the observation of high IFN-γ secretion from the lung cells of vaccinated young mice 48 h after stimulation with IpaB or IpaD, but the lack of significant upregulation of *Ifng* gene expression after vaccination or challenge. It is possible that *Ifng* gene expression occurred at a different time point, since IFN-γ secretion typically occurs later in the immune response cascade [47]. The gene expression of *Ifng* may have occurred before or after the time points used in our experiments. Moreover, IFN-γ is produced by various immune cells [48], so the observed secretion could be from immune cells, which are present at a level that is not easily detectable through standard gene expression analyses. Since the immune response is highly regulated, with multiple feedback mechanisms, the immune system on day 3 after *Shigella* infection may suppress the *Ifng* gene expression as a regulatory mechanism, or other cytokines interfered with its expression [48]. To gain a better understanding of the discrepancy between *Ifng* gene expression and protein secretion, more detailed research will be conducted in future work. 

## 5. Conclusions

In summary, the immune response in both young and elderly mice vaccinated with L-DBF/ME/BECC470 aids in effective pathogen clearance and resolution of infection, with a shift towards Th1 and Th17 responses. The statistically significant increase in IL-17 secretion in the lungs of both young and elderly mice, despite differences in immune function between age groups in the spleen, indicates that L-DBF/ME/BECC470 offers protection against *Shigella* infection in both, suggesting the potential benefits of enhancing immune function in elderly populations. These findings underscore age-related variations in immune responses to *Shigella* infection and emphasize the importance of tailoring vaccination formulations and strategies based on age-specific considerations.

## Figures and Tables

**Figure 1 vaccines-12-00618-f001:**
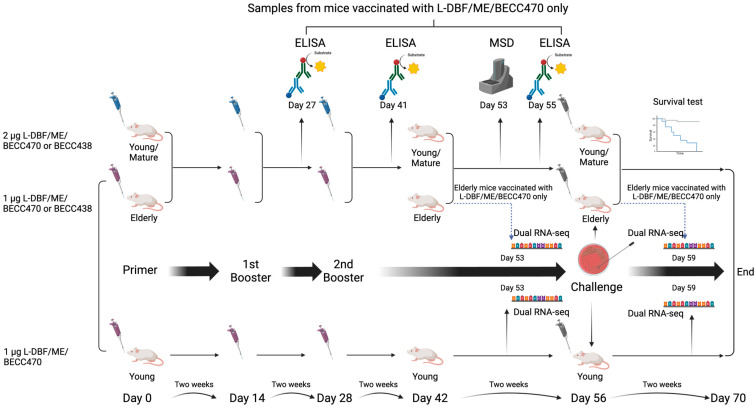
Experimental workflow for immunization and *Shigella* challenge studies. This schematic diagram illustrates the comprehensive experimental workflow used to study the immune response and bulk RNA-seq in mice. The procedure includes the following key steps: vaccination, immune response evaluation, *Shigella* lethal challenge, bulk RNA sequencing, and data analysis. Each section of the schematic is denoted by arrows, representing the flow and sequence of procedures. The color-coded elements indicate different experimental groups and types of analyses performed at each stage. Figure created using BioRender.com (accessed on 17 October 2023).

**Figure 2 vaccines-12-00618-f002:**
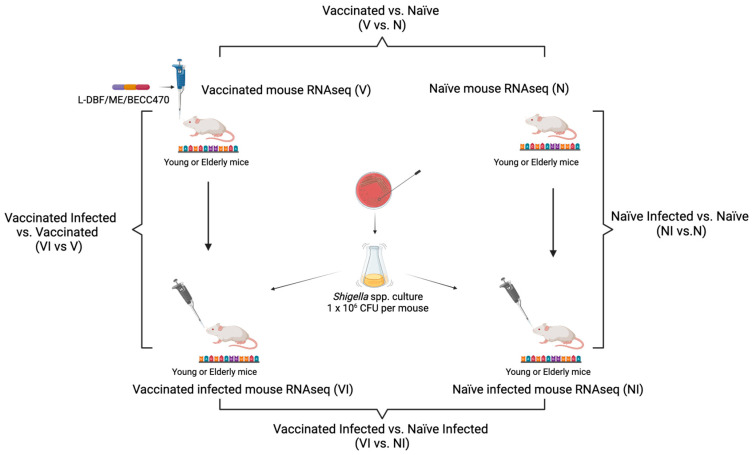
Experimental group comparisons for bulk RNA-seq analysis. Group pairs analyzed includes vaccinated mice vs. naïve mice (V vs. N), naïve infected mice vs. naïve mice (NI vs. N), vaccinated infected mice vs. vaccinated mice (VI vs. V), and vaccinated infected mice vs. naïve infected mice (VI vs. NI). Figure created with BioRender.com.

**Figure 3 vaccines-12-00618-f003:**
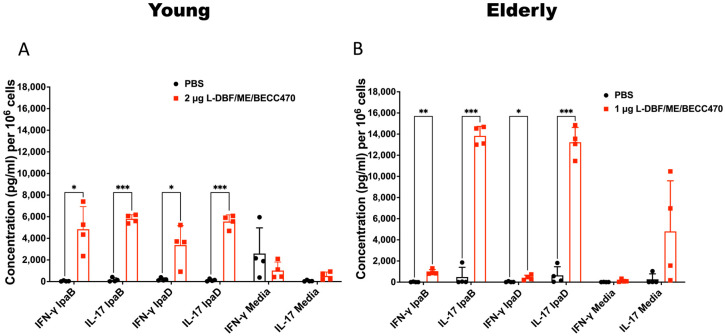
IFN-γ and IL-17A levels in lung cell suspensions collected from young (**A**) and elderly (**B**) mice vaccinated intranasally (IN) with PBS (Black) or L-DBF in BECC470/ME (Red). Single-cell lung suspensions were incubated with 10 µg IpaB and IpaD. Cytokine levels were determined by Meso Scale Discovery analysis as per the manufacturer’s specifications and are presented here as pg/mL/10^6^ cells. Secretion of different cytokines was noted as a response to either IpaB or IpaD stimulation. Data were plotted as actual values from individuals ± SD (n = 4) in each group. Significance was calculated by comparing groups that were unvaccinated (PBS) and mice vaccinated with antigens using a Welch *t*-test. * *p* < 0.05; ** *p*< 0.01; *** *p*< 0.001.

**Figure 4 vaccines-12-00618-f004:**
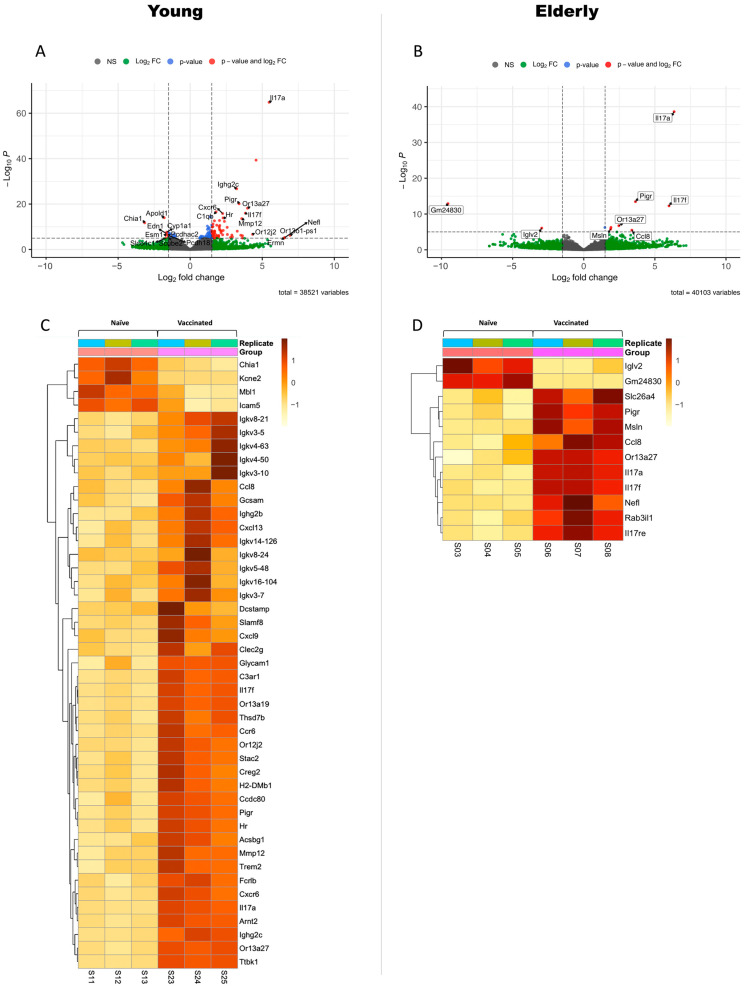
Differential gene expression in response to L-DBF/ME/BECC470 vaccination in different age groups. The top panels display a Volcano image illustrating the distribution of gene expression changes in young (**A**) or elderly (**B**) mice in response to 1 µg L-DBF/ME/BECC470. In both volcano images, gray spots indicate non-significant changes (NS), green spots represent |log2FoldChange| > 2 with *p* > 0.05, blue spots signify *p* < 0.05 and −2 < log2FoldChange < 2, while red spots highlight significantly changed genes with *p* < 0.05 and |log2FoldChange| > 2. Each area is separated by dash lines. Bottom panels feature heatmaps depicting gene expression changes in young (**C**) or elderly (**D**) mice responding to L-DBF/ME/BECC470 vaccination. The heatmaps provide a visual representation of genes with a significant differential expression, specifically those with *p* < 0.05 and |log2FoldChange| > 2. Each column corresponds to an individual sample, with the first three columns representing group N and the last three columns representing group V. The color scale reflects gene expression levels, where red hues indicate higher expression levels.

**Figure 5 vaccines-12-00618-f005:**
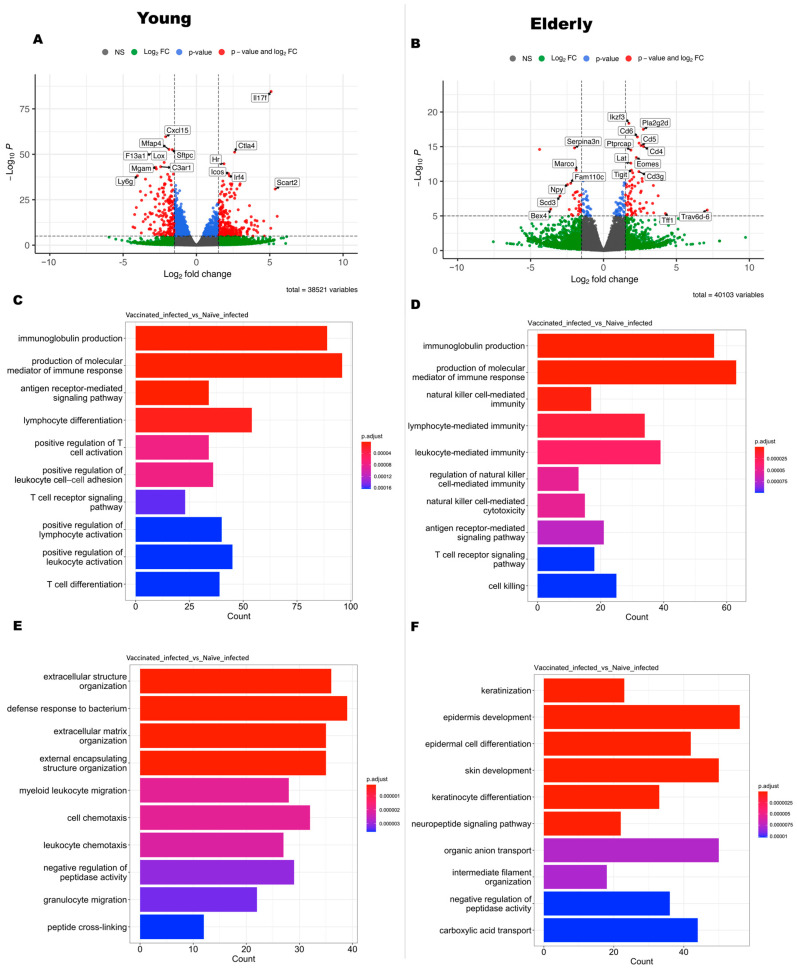
Gene expression changes and pathway analysis in response to *Shigella* spp. infection in vaccinated young (left) and elderly (right) mice (VI vs. NI). Top panels: Volcano images illustrating gene expression changes in VI young (**A**) or elderly (**B**) mice compared to their NI counterparts on day 3 after *Shigella* spp. infection. Gray spots denote non-significant changes (NS), green spots signify |log2FoldChange| > 2 with *p* > 0.05, blue spots represent *p* < 0.05 and −2 < log2FoldChange < 2, and red spots highlight significantly changed genes with *p* < 0.05 and |log2FoldChange| > 2. Each area is separated by dash lines. Middle panels: The top 10 upregulated pathways based on Gene Ontology (GO) enrichment analysis in young (**C**) or elderly (**D**) VI vs. NI comparison, offering insights into the biological processes associated with the upregulated genes. Bottom panels: The top 10 downregulated pathways based on GO enrichment analysis in young (**E**) or elderly (**F**) VI vs. NI comparison, shedding light on the biological processes affected by the downregulated genes.

**Figure 6 vaccines-12-00618-f006:**
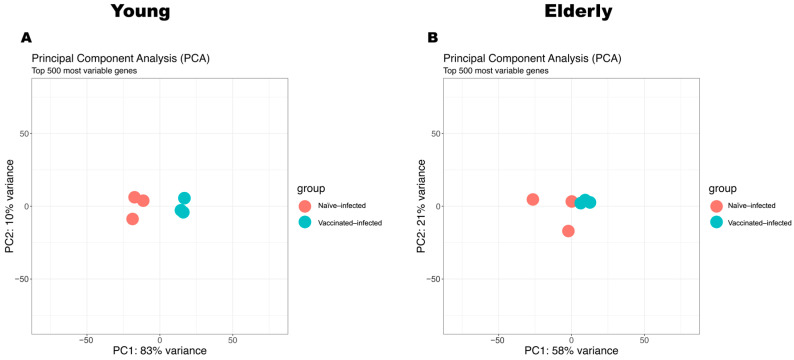
Principal component analysis (PCA) of transcriptomic profiles in response to *Shigella* infection post-vaccination (VI vs. NI). The PCA analysis incorporated the top 500 most variable genes in the young (**A**) or elderly (**B**) age group for a comprehensive assessment between NI (red dot) and VI (blue dot) mice.

**Figure 7 vaccines-12-00618-f007:**
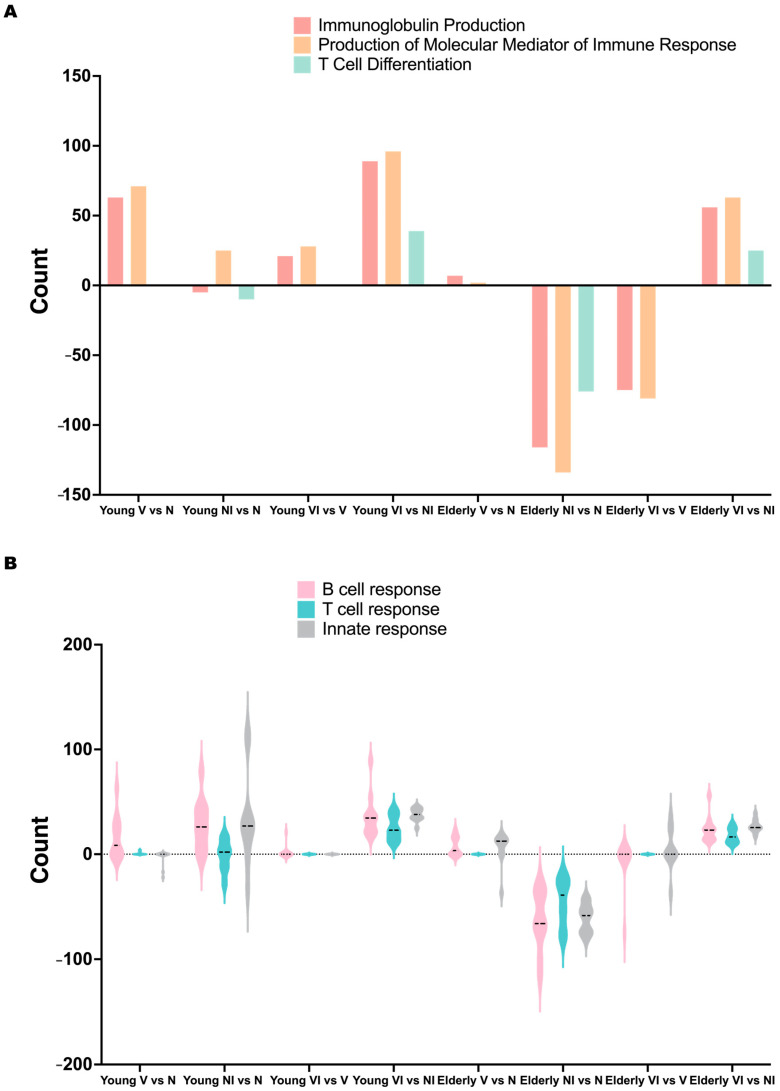
Immune pathway changes during vaccination and *Shigella* infection in naïve or vaccinated mice in young and elderly mice. Panel (**A**) shows the change in Gene Ontology (GO) enrichment count of immunoglobulin production (Red), production of molecular mediators of immune response (Yellow), and T cell differentiation (Green) upon vaccination and infection in young and elderly mice. The bar graph illustrates the alterations in the enrichment count of these key immune pathways across different experimental conditions, with y > 0 indicating upregulation and y < 0 indicating downregulation. Panel (**B**) shows the change in Gene Ontology (GO) enrichment count of different immune pathway clusters upon vaccination and infection in young and elderly mice. This bar graph presents the variations in the enrichment count of various immune pathway clusters (B cell response, Pink; T cell response, Blue; innate response, Gray) across different experimental conditions, with y > 0 indicating upregulation and y < 0 indicating downregulation.

**Figure 8 vaccines-12-00618-f008:**
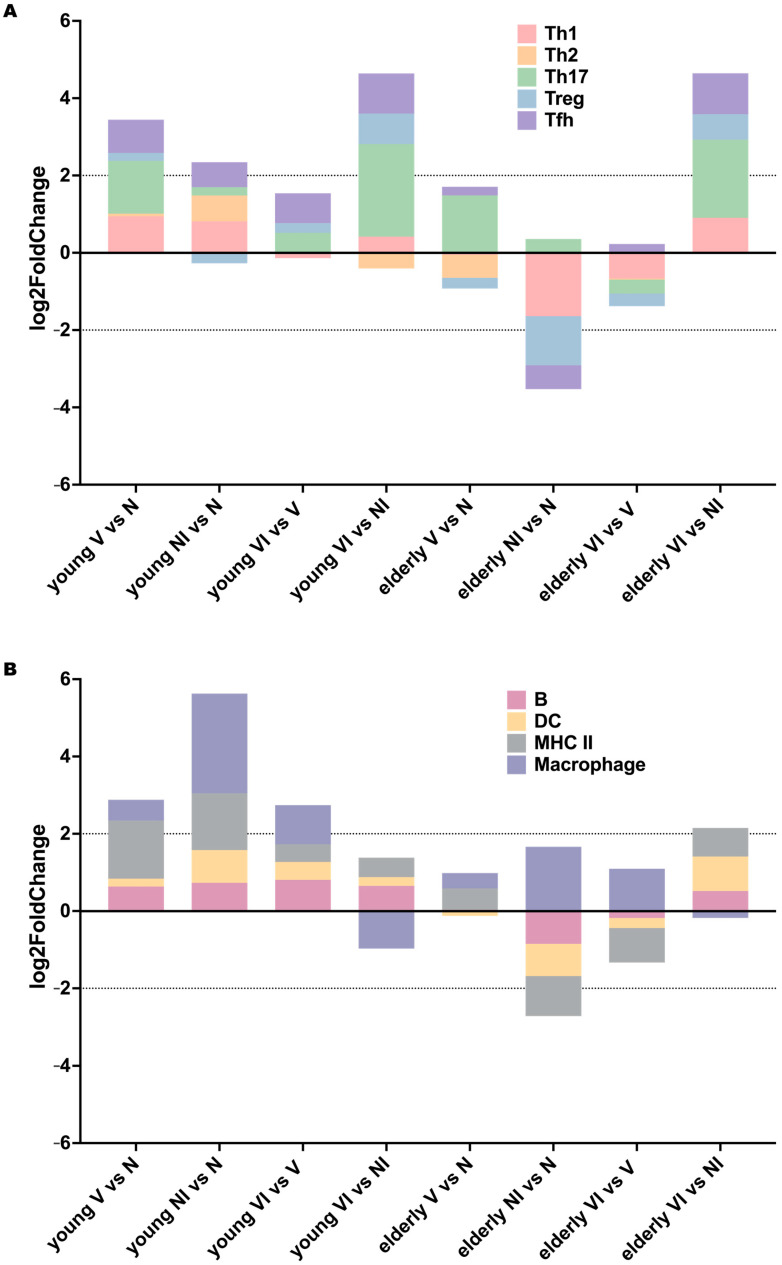
Total log2 fold change of genes in immune cell categories during vaccination and infection in young and elderly mice. Panel (**A**) presents the change in total log2 fold change of the Th1 (Red), Th2 (Yellow), Th17 (Green), Treg (Blue), and Tfh (Purple) genes during vaccination and infection in young and elderly mice. The stacked bar graph illustrates alterations in the mean of each category across different experimental conditions, with y > 0 indicating upregulation and y < 0 indicating downregulation. Panel (**B**) shows the change in total log2 fold change of B cells (Red), dendritic cells (Yellow), MHC II related genes (Gray), and macrophages (Purple) during vaccination and infection in young and elderly mice. The stacked bar graph illustrates alterations in the mean of each category across different experimental conditions, with y > 0 indicating upregulation and y < 0 indicating downregulation.

## Data Availability

The original contributions presented in the study are included in the article/Appendix A; further inquiries can be directed to the corresponding authors. The raw data supporting the conclusions of this article will be made available by the authors on request.

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
