# Peer review of "Vaccination with a Protective Ipa Protein-Containing Nanoemulsion Differentially Alters the Transcriptomic Profiles of Young and Elderly Mice following Shigella Infection"

_vaccines, 2024, doi:10.3390/vaccines12060618_

Round 1

Reviewer 1 Report

Comments and Suggestions for Authors

The manuscript presented describes vaccination with a water-in-oil emulsion protein vaccine against Shigella and transcriptomic differences as a function of age. The manuscript attempts to identify gene expression changes and pathway analyses in naive and vaccinated young vs old vaccinated species as well as following Shigella challenge. The authors utilize a plethora of sophisticated analytical techniques including ELISAs and RNAseq coupled with bioinformatic analyses to answer the proposed question. Overall, the manuscript is well-written and the analyses performed are very thorough. However, the following comments must be thoroughly addressed:

- The main issue with this work is the differences in dosing in the young and mature groups (2ug dose) compared to the old group (1ug dose). Such a stark difference of 100% in dose is not well justified in the manuscript and as a result some of the comparisons especially in magnitude of secreted cytokines and fold changes in gene expression cannot be compared across different aged groups due to the extreme difficulty in decoupling age and dose as two independent variables. The authors must thoroughly address this issue in the manuscript and make necessary changes to the results and discussion sections where direct comparisons are made between the two different age (and dosage) groups. The authors claim that the 2ug dose caused adverse events in the old group. What are those adverse events? Did the authors consider 1ug dose for all the groups? Is there data to justify the choice of dosage? The authors must address this major issue in the study design.

- Overall, the introduction is very well written with all the background described in great detail. One detail to add is an overview of the known age-related immune differences at the cell, gene, and protein levels.

-  In the old cohort, is the mouse age simply "older than eighteen months?" or is there variability in age within the cohort?

- How were lung cells isolated for ELISA and other analyses? Did the authors consider performing a bronchoalveolar lavage to look at IgA in the lung mucosa? IgA in the lung is a critical piece especially in an intranasal vaccine.

- The discussion around IL-17 ELISA results and the conclusions are not backed by the data as it is not statistically significant and there is large variability for IL-17. That language should be softened as that data does not conclusively prove Th17 responses. Also, Fig S4 shows the reverse observation for splenocytes.

- Overall, the authors should reconsider the cytokine ELISA results and focus on statistically significant events as the conclusions described in the results section are largely based on highly variable and statistically insignificant data. 

Author Response

Thank you very much for taking the time to review this manuscript. Please find the detailed responses in the attachment and the corrections highlighted in the re-submitted files

Reviewer 2 Report

Comments and Suggestions for Authors

Review of Manuscript on Shigella

1.  This is a complex, detailed, and interesting manuscript that examines the outcomes of vaccinating young (six-to-eight- week-old) and elderly (more than 18-month-old) mice against Shigella infection.  The vaccine used was a recombinant fusion protein made from the following:  two proteins of the type III secretion system of Shigella (IpaB and IpaD) plus a doubly mutated version of the A1 portion of the labile toxin of enterotoxic Escherichia coli, which the authors refer to as LTA1.  The recombinant fusion protein derived from these three proteins is called L-DBF.  In addition to L-DBF, the vaccine included one of two nontoxic lipid A analogs [Bacterial Enzymatic Combinatorial Chemistry candidate #348 (BECC438) or Bacterial Enzymatic Combinatorial Chemistry candidate #470 (BECC470).  Vaccine components (L-DBF+BECC438 or L-DBF+BECC470) were prepared in an oil in water emulsion and administered to mice intranasally.  Various responses of the mice to vaccination were examined and analyzed, such as cytokine production by lung cell suspensions or spleen cell suspensions, production of IgG and IgA, transcriptional responses (as determined by RNA sequencing) and responses to challenge with Shigella bacteria (including survival as well as transcriptional responses).  Naïve young and naïve elderly mice were included in these experiments as controls.  In a few experiments, mature (eight-to-eleven- month-old) mice were also examined.   

2.  This manuscript represents a great deal of work, and the findings are both intriguing and important, as there are currently no approved vaccines for Shigella.  However, the manuscript was difficult and time-consuming to read and review.  The reviewer respectfully offers the following comments for consideration by the authors. 

3. Line 84.  Perhaps a new paragraph should begin with this sentence: “In this study, …..”

4.  The Supplemental Table S1 (Acronyms used in this paper) was helpful.  However, this table should be expanded to include more of the abbreviations used in the manuscript (for example, IPTG, IMAC, MSD, DEG, GO, PCA, EU).  Although some of these abbreviations may be familiar to many readers, the manuscript will be more accessible if they are included in the table.

5.  Section 2.4 (Mice and Immunizations) of the Materials and Methods is not clear and should be revised.

6.  A descriptive legend is needed for Supplemental Figure S1 (Experimental workflow for immunization and Shigella challenge studies).  Perhaps this figure could be included in the manuscript rather than as a supplemental figure.

7.  Section 2.5.  Information should be given about how the fecal pellets were processed for the ELISA. 

8.  Line 296.  Replace the word “mouse” with “mice”.

9.  Supplemental Table S2 Description.

Current version: “Vaccine efficacy (VE) is presented as VE = 1 – Attack Rate Vaccinated/Attack Rate Unvaccinated (PBS control), wherein the control group receiving PBS vaccination observed mortality in all mice.”

Suggested revision: Vaccine efficacy (VE) is presented as VE = 1 – Attack Rate Vaccinated/Attack Rate Unvaccinated (PBS control).  All mice died in the control (PBS vaccinated) group.

10.  Supplemental Figure S4. Legend line 4.  The concentration should be indicated as 10 μg/ml.

11.  Supplemental Figure S5.  Panel B.  The label at the top of the figure should be fixed so it is in a format similar to the others: 2 μg L-DBF/ME/BECC470

12.  Supplemental Figure S5. Legend. 

Lines 8-10: Suggested revision: Cytokine levels in suspensions of lung cells (D & E) or splenocytes (F) collected on day 3 before the challenge of  mature mice are displayed.  Cell suspensions were incubated with 10 μg/ml IpaB and IpaD.

13.  Supplemental Figure S5. Legend.

Line 13.  The reviewer suggests replacing the word “of” with the word “to”.

14.  Supplemental Figure S6. Legend.

Lines 6: Suggested revision: The color intensity indicates the expression level, with heavier colors indicating higher (A) or lower (B) expression.

15.  Supplemental Figure S7. Legend.

Lines 2, 4, 10, & 14: Add the word “in” before the word “response”.

16.  Supplemental Figure S8. Legend.

Line 2.  Add the word “in” before the word “response”.

17.  Supplemental Figure S9. Legend.

Lines 3, 5, & 11, & 14.  Add the word “in” before the word “response”.

18.  Supplemental Figure S10. Legend.

Lines 3, 5, & 9. Add the word “in” before the word “response”.

19.  Supplemental Figure S11.  Legend.

Line 3. Add the word “in” before the word “response”.

20.  Line 312.  Delete the word “concentrate”.

21. Figure 2. Legend.

Line 315.  Suggested revision: IFN-g and IL-17A levels in lung cell suspensions collected………….

Line 316.  Suggested revision: Lung cell suspensions were incubated with 10 μg/ml IpaB and IpaD.

Line 318.  Fix exponent on cell number.

Line 319. Replace the word “of” with the word “to”.

22.  Line 339. Suggested revision: Replace the word “mice” with the word “mouse”.

23. Line 356. Suggested revision: Replace the words “in the lungs” with the words “in lung cell suspensions”.

24.  Lines 352-360.  This paragraph would be clearer if additional references to the appropriate figures were included.  For example, lines 352-353 could be revised as follows:

However, the degree of weight loss experienced was less pronounced (Figure S5B) compared to the elderly group (Figure XXX).  Increases in anti-IpaD and -IpaB IgG and IgA were observed following vaccination in mature groups (Figure S5C), showing levels closely resembling those observed in the young group (Figure XXXX).  Heightened secretion of IFN-g and IL-17 was detected in lung cell suspensions from vaccinated mature mice (Figure S5D), a trend mirroring that observed in the young mice (Figure XXXXX). Notably, IL-6 levels in lung cell suspensions from mature mice increased upon media stimulation, and TNF-α levels rose after IpaB stimulation (Figure S5E).  However, in mature mice, only IL-6 levels in cultured splenocytes were significantly elevated after IpaD stimulation compared with the levels in the control cultures.  (The reviewer did not identify the appropriate figures and used the letter(s) “X” in place of the numbers.)

25.  Line 371. Insert the word “on” after the word “focused”.

26.  Line 374.  Suggested revision: (day 53 of total workflow; V vs. N, Figure 1 and Figure S1).

27.  Lines 376-377.  If the changes noted are significant, shouldn’t it be p < 0.05?

28.  Line 387-388.  If the changes noted are significant, shouldn’t it be p < 0.05?

29.  Figure 3A is difficult to observe because of the scale, overlapping data points, and the rectangular gene labels.  The authors refer to Figure 3A and state that transcription of 248 genes was significantly increased in the young mice in response to vaccination (lines 376-377).  The reviewer has no way to assess the validity of this statement.  Not all upregulated genes are shown in Figure S6A. 

30.  Figure 3 legend.  Lines 403-406.  These statements are not clear.

31.  Line 421. Suggested revision: “highlights” rather than “highlight”.

32.  Line 483. Suggested revision: Replace the word “more” with the word “greater”.

33.  Line 501. Suggested revision: Replace the word “are” with the word “were”.

34.  Lines 565-566. Suggested revision: The top downregulated pathways found in young VI mice compared to young NI mice were (Figure 4E): extracellular structure organization; defense response to bacterium;…..

35.  Line 600. Suggested revision: Replace the word “included” with the word “including”.

36.  Line 625. Suggested revision: Replace the word “more” with the word “greater”.

37.  Figure 6 and lines 662-667.  The V versus N comparisons for both young mice and elderly mice should be added in both Figure 6A and Figure 6B. Without these comparisons, it is difficult to properly evaluate the conclusions in lines 662-667.

38.  Line s 646-647.  Suggested revision: Replace the word “means” with the word “indicating” in both lines.

39.  Line 736.  Suggested revision: Possibly change “The absence of such upregulation…” to “The greatly diminished upregulation…”

40.  Lines 744-750.  While it seems likely that the Th-17 responses are important, can one know from this study that they are responsible for the observed vaccine-mediated protection in the elderly mice without conducting additional experiments?  Use of deficient mouse strains or depletion of specific cell types or cytokines might be useful to establish the role of the Th-17 responses.  Would you expect transfer of Th-17 cells from immunized mice to provide protection to naïve mice?

41.  Lines 759-796.  This paragraph is rather long and could be better organized.

Comments on the Quality of English Language

I prepared only one list of comments.  Time limitations prevented me from separating the comments related to the quality of the English language from those related to the study itself.

Author Response

Thank you very much for taking the time to review this manuscript. Please find the detailed responses in the attachment and the corrections highlighted in the re-submitted files.

Round 2

Reviewer 1 Report

Comments and Suggestions for Authors

I thank the authors for the detailed response and revisions to the manuscript. The responses satisfy the questions.